# A Latent Causal Diffusion Model for Multi-objective Reinforcement Learning

## Abstract

Multiobjective reinforcement learning (MORL) poses significant challenges due to the inherent conflicts between objectives and the difficulty of adapting to dynamic environments. Traditional methods often struggle to generalize effectively, particularly in large and complex state-action spaces. To address these limitations, we introduce the Latent Causal Diffusion Model (LacaDM), a novel approach designed to enhance the adaptability of MORL in discrete and continuous environments. Unlike existing methods that primarily address conflicts between objectives, LacaDM learns latent temporal causal relationships between environmental states and policies, enabling efficient knowledge transfer across diverse MORL scenarios. By embedding these causal structures within a diffusion model-based framework, LacaDM achieves a balance between conflicting objectives while maintaining strong generalization capabilities in previously unseen environments. Empirical evaluations on various tasks from the MOGymnasium framework demonstrate that LacaDM consistently outperforms the state-of-art baselines in terms of hypervolume, sparsity, and expected utility maximization, showcasing its effectiveness in complex multiobjective tasks.

## 1 Introduction

Multiobjective reinforcement learning (MORL) Felten et al. (2024a; 2023a); Hayes et al. (2022); Yang et al. (2019); Lu et al. (2023) has become a prominent research area due to its ability to address real-world problems where multiple, often conflicting objectives must be optimized simultaneously. In contrast to traditional reinforcement learning, where a single objective is maximized, MORL requires to balance and optimize multiple objectives, often in dynamic and uncertain environments Gu et al. (2025). This adds significant complexity to the learning process, especially when the objectives conflict with each other or change over time. Moreover, large, high-dimensional state-action spaces in many MORL tasks pose scalability challenges to existing algorithms, making it difficult for MORL to generalize across different tasks Felten et al. (2024b).

Traditional approaches to MORL have primarily focused on scalarization methods Van Moffaert et al. (2013); Zheng & Wang (2023), Pareto-based methods Cai et al. (2023); Liu et al. (2025a); Van Moffaert & Nowé (2014); Zheng & Wang (2022). While these methods show impressive success in solving multiobjective problems, they often struggle to generalize, particularly in complex environments where the relationships between objectives are nonlinear and the state-action space are subject to changes. Scalarization methods Agarwal et al. (2022); Gu et al. (2024) combine objectives into a weighted sum or other forms of aggregation, but the choice of weights often requires prior knowledge of the relative importance of each objective, which may not be available in real-world scenarios Xia et al. (2021). Pareto-based methods Tian et al. (2022), which rank solutions based on their dominance relationships, often require to maintain a set of nondominated solutions, which can be computationally expensive, especially in high-dimensional spaces. Furthermore, Pareto-based approaches struggle when objectives conflict in complex ways or when the set of nondominated solutions becomes large and difficult to manage Lin et al. (2022). Additionally, both scalarization and Pareto-based methods Zhang et al. (2024b) typically assume that the objectives remain relatively stationary, which is not always the case in dynamic or uncertain environments. Therefore, these methods often struggle with generalization, particularly in complex environments where the relationships between the objectives are nonlinear, and the state-action space is highly time-varying.

Recently, diffusion models Ho et al. (2020); Sanokowski et al. (2024); Song et al. (2020) have demonstrated attractive generalization abilities and been applied to various types of optimization problems. For example, EmoDM Yan & Jin (2024) adopts a diffusion model to solve multi-objective optimization problems (MOPs) that considers the evolutionary search process as a reverse diffusion process. By pretraining on previously solved MOPs, EmoDM can generate a set of nondominated solutions for a new MOP through reverse diffusion, without the need to perform additional evolutionary search. A generative diffusion model, called GDMTD3 Zhang et al. (2024a), was proposed to solve the aerial collaborative secure communication optimization problem in multiobjective reinforcement learning. While these diffusion models have advanced the field of multiobjective optimization Li et al. (2024); Zhang et al. (2024a), they still require substantial training on past optimization tasks to generalize effectively. This reliance on extensive pretraining limits the model's adaptability to new, unseen environments, hindering its scalability and applicability to complex, real-world MORL problems Zeng et al. (2024). Furthermore, most of these diffusion models fail to account for the temporal and latent dependencies that arise from the interaction between the agent's actions and the evolving environment, which are critical for robust decision-making in dynamic settings and also significantly impact generalization performance Zhang et al. (2024a).

Fortunately, we find that the interactive mechanism between the agent and the environment can be characterized by causality. Causal inference is a useful tool for modeling the generative mechanism behind the agent's actions. Inspired by this, we propose a Latent Causal Diffusion Model (LacaDM) for MORL, which incorporates causal relationships between objectives and the environment directly into its latent space, enabling more structured and data-efficient policy generation. In the forward diffusion process, LacaDM learns to approximate the reserved search process of a deep reinforcement learning algorithm, such as Pareto Conditioned Networks (PCN) Reymond et al. (2022), starting from the optimal policy and gradually evolving toward a random initial policy. In the reverse diffusion process, noise is progressively removed from the random policy, incrementally refining it into an optimal policy approximation. The main contributions of this work are as follows:

- We propose a latent causal diffusion model to solve MORL problems. The key idea is to integrate causal representation learning into the diffusion process to enhance generalization across a wide range of MORL tasks. LacaDM can be used as a general diffusion model applicable to both continuous and discrete MORL environments.

- We optimize the policy by learning the latent causal dynamics during the reverse diffusion process, thereby improving LacaDM's ability to adapt to dynamic environments. This strategy enables LacaDM to continuously learn and adapt to new tasks without the requirement for exhaustive pretraining on previous problems.

- Extensive experiments conducted across a variety of environments from MOGymnasium demonstrate the superiority of LacaDM over existing baseline methods in terms of hypervolume, sparsity and expected utility maximization across a wide range of MORL tasks.

## 2 PRELIMINARIES

### 2.1 DIFFUSION PROBABILISTIC MODELS

Diffusion models Yang et al. (2023) are generative frameworks that refine noise-corrupted samples to generate high-quality data. They consist of two phases: the forward process, which adds noise, and the reverse process, which removes it to recover the original sample. These models Nie et al. (2025); Guo et al. (2024); Ye et al. (2022) are classified into two types: continuous diffusion models, for continuous data (e.g., images, audio), and discrete diffusion models, for discrete data (e.g., binary, categorical). While both follow similar principles, they differ in how noise is added and removed.

**Continuous diffusion models.** For continuous data, the forward process corrupts the initial sample $\mathbf{x}_0$ by adding Gaussian noise over $T$ steps, forming a Markov chain $\{\mathbf{x}_t\}_{t=0}^{T}$. Each step introduces noise based on a variance schedule $\beta_t$:

$$q(\mathbf{x}_t \mid \mathbf{x}_{t-1}) = \mathcal{N}(\mathbf{x}_t; \sqrt{1-\beta_t} \cdot \mathbf{x}_{t-1}, \beta_t \cdot \mathbf{I}), \tag{1}$$

where $\beta_t \in (0, 1)$. As $t$ increases, the sample becomes increasingly indistinguishable from pure noise. The noise is gradually accumulated, and we can express the relation between $\mathbf{x}_t$ and the

original sample $\mathbf{x}_0$ as:

$$q(\mathbf{x}_t \mid \mathbf{x}_0) = \mathcal{N}(\mathbf{x}_t; \sqrt{\bar{\alpha}_t} \cdot \mathbf{x}_0, (1 - \bar{\alpha}_t) \cdot \mathbf{I}), \tag{2}$$

where $\bar{\alpha}_t = \prod_{s=1}^{t}(1 - \beta_s)$ represents the cumulative noise added up to time $t$. This relation shows that as the forward process progresses, the data becomes more corrupted, with $\bar{\alpha}_t$ diminishing over time. The reverse process aims to denoise the final noisy sample $\mathbf{x}_T$ and recover the original sample $\mathbf{x}_0$. This process learns a parameterized distribution:

$$p_\theta(\mathbf{x}_{t-1} \mid \mathbf{x}_t), \tag{3}$$

which is typically modeled as a Gaussian distribution with learnable mean $\mu_\theta$ and variance $\Sigma_\theta$:

$$p_\theta(\mathbf{x}_{t-1} \mid \mathbf{x}_t) = \mathcal{N}(\mathbf{x}_{t-1}; \mu_\theta(\mathbf{x}_t, t), \Sigma_\theta(\mathbf{x}_t, t)). \tag{4}$$

During training, $\mu_\theta$ is optimized to effectively remove noise at each step, while $\Sigma_\theta$ is often fixed or simplified for stability. By iteratively "denoising" $\mathbf{x}_t$, the model can recover the original data distribution.

**Discrete diffusion models.** For discrete data, the forward process corrupts the initial sample $\mathbf{x}_0$ using Bernoulli noise over $T$ steps, forming a Markov chain $\{\mathbf{x}_t\}_{t=0}^{T}$. Each step flips the values of the data with a probability $\beta_t$:

$$q(\mathbf{x}_t \mid \mathbf{x}_{t-1}) = \text{Bernoulli}\left(\mathbf{x}_t; (1 - \beta_t)\mathbf{x}_{t-1} + \frac{\beta_t}{2}\right). \tag{5}$$

As $t$ increases, $\mathbf{x}_t$ becomes increasingly uniform over the discrete space. The distribution of $\mathbf{x}_t$ is gradually smoothed, with the relationship between $\mathbf{x}_t$ and $\mathbf{x}_0$ given by:

$$q(\mathbf{x}_t \mid \mathbf{x}_0) = \text{Bernoulli}\left(\mathbf{x}_t; \gamma_t \cdot \mathbf{x}_0 + \frac{1 - \gamma_t}{2}\right), \tag{6}$$

where $\gamma_t = \prod_{s=1}^{t}(1 - \beta_s)$. As $\gamma_t \to 0$, $\mathbf{x}_t$ approaches a uniform distribution over the discrete space. The reverse process seeks to denoise $\mathbf{x}_T$ back to $\mathbf{x}_0$ by learning the distribution:

$$p_\theta(\mathbf{x}_{t-1} \mid \mathbf{x}_t), \tag{7}$$

which is parameterized as a discrete distribution:

$$p_\theta(\mathbf{x}_{t-1} \mid \mathbf{x}_t) = \text{Bernoulli}\left(\mathbf{x}_{t-1}; f_\theta(\mathbf{x}_t, t)\right). \tag{8}$$

During training, the model minimizes the cross-entropy loss between the predicted distribution $p_\theta$ and the true posterior $q(\mathbf{x}_{t-1} \mid \mathbf{x}_t, \mathbf{x}_0)$, analogous to noise prediction in continuous diffusion models.

## 2.2 MULTIOBJECTIVE REINFORCEMENT LEARNING

In many real world scenarios, decision-making involves optimizing multiple, often conflicting objectives. Multiobjective reinforcement learning (MORL) Hayes et al. (2022) extends the standard RL framework by utilizing a vector-valued reward signal

$$\mathbf{r}_t = [r_t^{(1)}, r_t^{(2)}, \ldots, r_t^{(m)}], \tag{9}$$

where each component $r_t^{(i)}$ corresponds to a distinct objective. Improving one objective can degrade performance in another, so specialized algorithms are required to handle these trade-offs effectively.

Formally, an MORL environment is typically represented by a Markov Decision Process (MDP) $\langle \mathcal{S}, \mathcal{A}, \mathcal{P}, \mathbf{r}, \gamma \rangle$. Here, $\mathcal{S}$ is the state space, $\mathcal{A}$ is the action space, $\mathcal{P}(s' \mid s, a)$ defines the transition probabilities, and $\gamma \in [0, 1)$ is the discount factor. The vector reward function $\mathbf{r} : \mathcal{S} \times \mathcal{A} \to \mathbb{R}^m$ provides different reward signals for each objective. The goal is to learn a policy $\pi : \mathcal{S} \to \mathcal{A}$ that balances multiple objectives as encoded in the vector-valued return:

$$\mathbf{G}_t = \sum_{k=0}^{\infty} \gamma^k \mathbf{r}_{t+k}, \tag{10}$$

where $\mathbf{G}_t = [G_t^{(1)}, G_t^{(2)}, \ldots, G_t^{(m)}]$ accumulates the rewards for each objective. Because no single solution can optimize all objectives simultaneously, *Pareto optimality* is often used to evaluate policies that cannot be improved in one objective without sacrificing another. Managing these high-dimensional, conflicting objectives remains a significant challenge. In MORL Liu et al. (2025b); Zhu et al. (2023), an agent must handle multiple, often conflicting objectives which can be viewed as different dimensions or "channels" of a complex decision space. By leveraging diffusion-based methods, the proposed LacaDM can generate candidate policies that systematically explore and refine this high-dimensional space, potentially yielding a diverse set of Pareto optimal solutions.

## 3 CAUSAL REPRESENTATION LEARNING FOR MORL

In nonstationary environments, reward dynamics and objective trade-offs may shift over time due to latent factors, making it increasingly challenging for the agent to balance multiple, often conflicting objectives. Effectively modeling the underlying causal structure over these latent factors is essential for achieving robust, generalizable, and adaptive policy behavior.

**Temporally causal representation modeling.** We model the causal representation based on latent temporally causal processes Yao et al. (2021), which aims to recover latent factors driving temporal dynamics. Formally, let $\mathbf{x}_t$ denote the observed state-action-reward tuple at time $t$. The data generating mechanism can be modeled as:

$$\mathbf{x}_t = g(\mathbf{z}_t), \quad \mathbf{z}_t \in \mathbb{R}^k \tag{11}$$

where $\mathbf{z}_t = (z_{1,t}, \ldots, z_{k,t})$ are latent variables evolving via a delayed causal process:

$$z_{i,t} = f_i\left(\{\mathbf{z}_{t-\tau}\}_{\tau=1}^L, \epsilon_{i,t}\right), \quad \forall i \in \{1, \ldots, k\} \tag{12}$$

Here, $f_i$ captures nonlinear causal influences from past latent states, and $\epsilon_{i,t}$ represents exogenous noise due to unobserved environment shifts. This is corresponded to a temporal causal structure: $\{\mathbf{z}_{t-\tau}\}_{\tau=1}^L \rightarrow \mathbf{z}_t \rightarrow \mathbf{x}_t$. Under this model, $\mathbf{z}_t$ encodes latent task preferences and environmental dynamics influencing observed behaviors.

**Policy adaptation via causal inference.** To enable policy adaptation, we use an encoder-decoder architecture to infer latent variables $\mathbf{z}_t$ and model their dynamics. When a shift $\Delta z_{i,t}$ is detected due to environmental perturbations, the policy input is adjusted proactively:

$$a_t = \pi_\theta\left(\{\mathbf{z}_{t-\tau}\}_{\tau=1}^L + \Delta z_{i,t}\right). \tag{13}$$

This allows counterfactual reasoning, i.e., simulating actions under altered causal contexts.

**Agent and environment effects disentanglement.** To separate external environment drift from internal policy effects, we model the conditional distribution of the noise term:

$$p(\epsilon_{i,t}) = \frac{\partial s_i(\epsilon_{i,t})}{\partial \epsilon_{i,t}} \cdot \mathcal{N}(s_i(\epsilon_{i,t})), \tag{14}$$

where $s_i(\cdot)$ is a normalizing flow transforming $\epsilon_{i,t}$ into a standard distribution. This helps identify whether observed deviations stem from environmental shifts or policy deficiencies.

In summary, causal representation learning (CRL) equips the MORL agent with essential reasoning capabilities that are difficult to achieve with conventional methods. These include the ability to intervene on latent task or environmental factors to synthesize adaptive policies, reason counterfactually about hypothetical situations, and model temporally delayed causal effects in dynamic environments. Together, these capabilities provide a principled and actionable foundation for enhancing generalization, robustness, and adaptability in multi-objective reinforcement learning.

## 4 METHODOLOGY

In this section, we first introduce the overview of the proposed LacaDM, and then detail the key components of the LacaDM framework, including forward diffusion for noise estimation, and generation of optimal policies via reverse diffusion.

### 4.1 OVERVIEW

Figure 1 illustrates the overall architecture of the proposed LacaDM, which integrates latent causal modeling with a bidirectional diffusion process to enable robust policy learning in MORL environments. The framework comprises a forward diffusion that progressively injects noise into the policy space to promote exploration and diversity, and a reverse diffusion that iteratively removes noise to recover high-quality policies. To guide the forward diffusion during action generation, we construct an inverse reinforcement learning (IRL) context embedding with historical trajectories from PCN Reymond et al. (2022); Beliaev & Pedarsani (2025). This embedding captures temporal dependencies within state-action-reward sequences and provides a compact summary of the agent's

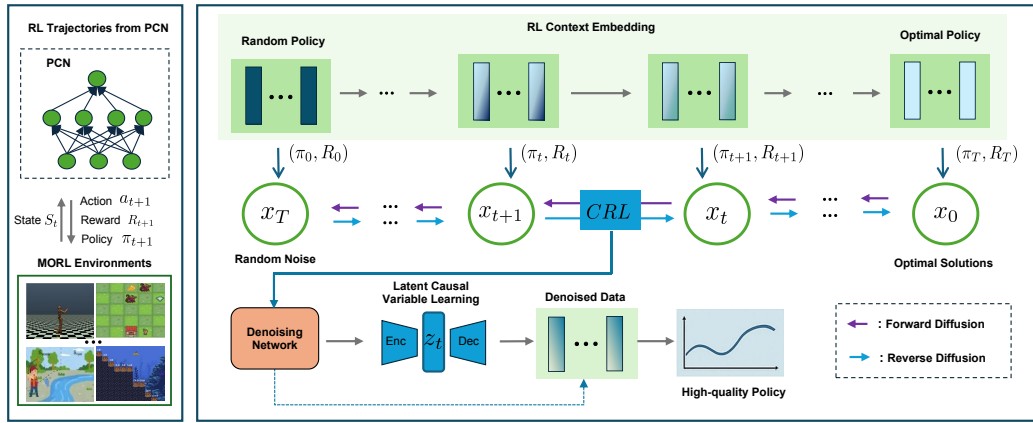

Figure 1: Overview of the proposed LacaDM. Inverse RL-guided context embeddings derived from PCN-generated trajectories guide the forward diffusion from optimal solutions to random noise. Latent causal variables $z_t$, learned via CRL, support reverse denoising to recover high-quality policy solutions.

past behaviors under varying objective trade-offs. CRL serves different purposes across the two processes: during forward diffusion, it extracts latent causal variables from observed trajectories, while during reverse diffusion, these variables are used to guide denoising and policy reconstruction. By modeling causal relationships between extracted latent causal variables, CRL improves LaCaDM's ability to generalize across nonstationary tasks with shifting preferences and dynamics.

## 4.2 Forward Diffusion for Noise Estimation

In the proposed LacaDM, we use a reinforcement learning context embedding to guide the diffusion process, as illustrated in Fig. 1. Specifically, the policy at each time step, along with the changes in cumulative rewards, serves as a reinforcement signal that helps LacaDM capture the implicit optimization behavior of different RL algorithms. We begin by solving $N$ distinct MORL problems to convergence and recording the resulting policy $\pi_T$ and cumulative reward sequences $R_T$:

$$\pi_T = \{\pi_1, \pi_2, \dots, \pi_t\}, \quad R_T = \{R_1, R_2, \dots, R_t\}, \quad (15)$$

where $\pi_t = \{\pi_t(s_1), \pi_t(s_2), \dots, \pi_t(s_n)\}$ denotes the policy at time step $t$, and $T$ is the total number of time steps used in the diffusion process. These sequences form the conditioning input to our diffusion model.

The forward diffusion process progressively estimates a noise distribution from the policy sequence $\pi_T$, refining it to match a predefined target distribution. We use a Gaussian noise model in continuous action spaces and a Bernoulli model in discrete environments. At each step $t$, the noise is injected and updated depending on the environment characteristics, starting from an initial time step $t = 0$. To go beyond pure stochastic degradation and capture environment-specific structure in policy evolution, we introduce a sequence of latent variables $\{z_t\}$, inferred from the joint trajectories of $\pi_t$ and $R_t$. These variables follow a causal generative process:

$$z_t = f(Pa(z_t), \varepsilon_t), \quad (16)$$

where $Pa(z_t)$ are the causal parents from previous steps and $\varepsilon_t$ represents diffusion noise. These latent variables capture both environment-specific semantics and temporal dependencies. An encoder is trained to infer these latent embeddings based on CRL, allowing LacaDM to model not only noise but also the underlying causal dynamics of policy transitions.

This iterative procedure continues until the noise model converges. The outcome is a pretrained LacaDM model that has internalized both the stochastic degradation and the latent causal structure of MORL processes, enabling effective policy generation in the reverse diffusion phase.

### 4.3 GENERATION OF OPTIMAL POLICIES VIA REVERSE DIFFUSION

The reverse diffusion process in LacaDM aims to recover the optimal policy $\hat{\pi}_0$ from noise-corrupted inputs by progressively denoising a sequence of latent policy representations $\hat{\pi}_t$. This process is governed by the following transition model:

$$p(\hat{\pi}_t|\hat{\pi}_{t-1}) = \begin{cases} \mathcal{N}(\hat{\pi}_t; \mu(\hat{\pi}_{t-1}), \sigma^2), & \text{if continuous} \\ \text{Bernoulli}(\hat{\pi}_t; \pi(\hat{\pi}_{t-1})), & \text{if discrete} \end{cases} \tag{17}$$

Here, $\mu(\cdot)$, $\sigma^2$, and $\pi(\cdot)$ are predicted by a denoising network, modeling the policy transition dynamics for both continuous and discrete environments. The process proceeds iteratively until a denoised policy $\hat{\pi}_0$ is obtained.

**Incorporating causal learning.** To enhance adaptability across diverse MORL environments, we introduce CRL in our LacaDM. At each reverse diffusion step, the denoised policy $\hat{\pi}_t$ is not only conditioned on its previous state $\hat{\pi}_{t-1}$, but also influenced by a latent variable $z_t$ that encodes underlying causal structure, which can be formalized as

$$\hat{\pi}_t = f(\hat{\pi}_{t-1}, z_t, \varepsilon_{\hat{\pi}_t}), \tag{18}$$

where $\varepsilon_{\hat{\pi}_t}$ is stochastic noise from the forward diffusion process.

Based on Eq. (12) and Eq. (18), an encoder network is trained to infer $z_t$ from observed noisy policy trajectories. It maps local segments of policy history to compact latent embeddings, which then guide the reverse diffusion process. The paired decoder $d(z_t, \hat{\pi}_{t-1})$ predicts the denoised policy step, enforcing consistency between causal latent variables and policy evolution. This mechanism enables LacaDM to model not just statistical transitions, but also the structural and temporal causal dynamics behind policy evolution.

**CRL-guided policy update.** Rather than treating reverse diffusion purely as a sampling procedure, we interpret each step as a local policy optimization guided by both denoising accuracy and causal coherence. Specifically, we define a composite loss:

$$\mathcal{L}_{\text{total}}(\hat{\pi}_{t-1}, \hat{\pi}_t, z_t) = \underbrace{\|\mu_\theta(\hat{\pi}_t) - \hat{\pi}_{t-1}\|^2}_{\text{denoising loss}} \\ + \beta \cdot \underbrace{\|d(\hat{\pi}_{t-1}, z_t) - \hat{\pi}_t\|^2}_{\text{causal consistency}} + \lambda \cdot \|\hat{\pi}_{t-1}\|_1. \tag{19}$$

The first term ensures accurate denoising from the noisy trajectory, the second term enforces consistency with the latent causal dynamics via a learned decoder $d$, and the third introduces $L_1$ regularization to encourage sparsity for improved generalization. We then update the policy using gradient descent:

$$\hat{\pi}_{t-1}^{\text{CRL}} = \hat{\pi}_{t-1} - \alpha \cdot \nabla_{\hat{\pi}_{t-1}} \mathcal{L}_{\text{total}}(\hat{\pi}_{t-1}, \hat{\pi}_t, z_t). \tag{20}$$

By embedding this loss-guided update into each reverse diffusion step, LacaDM refines policies not only based on statistical reconstruction but also on latent causal structure. This hybrid learning mechanism improves robustness and adaptability, particularly in scenarios involving domain shifts or temporally evolving objectives.

## 5 EXPERIMENTS

### 5.1 EXPERIMENT SETUP

**MORL environments.** To evaluate the performance of LacaDM, we adopt the MOGymnasium framework Felten et al. (2023b), a standardized benchmark suite for MORL. Built on Gymnasium, it supports a wide range of environments with multiple objective functions. We select eight discrete and eight continuous environments from MOGymnasium. The discrete environments include Deep Sea Treasure, HighwayEnv, ResourceGathering, FourRoom, FruitTree, Breakable Bottles, Fishwood, and MOLunarLander. The continuous environments include MountainCar, WaterReservoir,

HopperEnv, MOHalfCheetah, MOAnt, MOSwimmer, MOHumanoid, and MOWalker2D. This diverse selection enables a comprehensive evaluation of LaCaDM's performance across both discrete and continuous MORL tasks.

**Training datasets and Baselines.** To construct the training dataset for LacaDM, we use the PCN Reymond et al. (2022) to solve four MORL environments: Minecart, MOSuperMario, MOReacher, and DeepSeaTreasureMirrored. During agent-environment interaction, we record the state, action, reward, and policy at each time step, resulting in a rich trajectory dataset that captures the temporal dynamics and optimization behavior of multiobjective tasks. This dataset is used to supervise the forward diffusion process in LaCaDM, enabling the model to capture underlying patterns in policy evolution and multiobjective decision-making.

We compare LacaDM against a diverse set of baselines across both continuous and discrete MORL tasks, using their default hyperparameters. The baselines include two reinforcement learning methods (DQN Mnih (2013), PCN Reymond et al. (2022)), two evolutionary algorithms (NSGA-III-EHVI Pang et al. (2022), ANSGA-II Liu et al. (2022)), and three diffusion-based models (EmoDM Yan & Jin (2024), MTDiff He et al. (2023), DMBP Zhihe & Xu (2023)). These baselines cover traditional, evolutionary, and generative approaches, offering a comprehensive benchmark for evaluating LacaDM.

Table 1: Comparison of average HV results in discrete and continuous MORL environments.

| MORL Environment | Env. Type | Deep Qlearning | PCN | ANSGAII | NSGAIIIEHVI | EmoDM | MTDiff | DMBP | LacaDM (Ours) | p-value |
|---|---|---|---|---|---|---|---|---|---|---|
| Deep Sea Treasure | Discrete | 2.63e+2 | 3.02e+2 | 2.56e+2 | 3.34e+2 | 2.63e+2 | 3.48e+2 | **3.52e+2** | 3.50e+2 | 0.000 |
| HighwayEnv | Discrete | 1.08e+4 | 1.31e+4 | 9.84e+3 | 2.40e+4 | 9.90e+3 | 2.49e+4 | 2.44e+4 | **2.51e+4** | 0.000 |
| ResourceGathering | Discrete | 3.83e+0 | 4.74e+0 | 3.94e+0 | 4.70e+0 | 3.95e+0 | 4.80e+0 | 4.81e+0 | **4.82e+0** | 0.003 |
| FourRoom | Discrete | 2.13e+1 | 2.25e+1 | 2.24e+1 | 2.56e+1 | 2.18e+1 | 2.55e+1 | **2.72e+1** | 2.60e+1 | 0.000 |
| FruitTree | Discrete | 3.34e+4 | 3.31e+4 | 2.98e+4 | 3.45e+4 | 2.84e+4 | 3.57e+4 | 3.61e+4 | **3.64e+4** | 0.000 |
| BreakableBottles | Discrete | 2.54e+4 | 2.34e+4 | 2.65e+4 | 2.81e+4 | 2.67e+4 | 2.81e+4 | 2.79e+4 | **2.82e+4** | 0.100 |
| Fishwood | Discrete | 3.12e+3 | 3.03e+3 | 2.84e+3 | 3.02e+3 | 2.98e+3 | **3.17e+3** | 3.05e+3 | 3.15e+3 | 0.000 |
| MOLunarLander | Discrete | 8.21e+8 | 8.15e+8 | 8.10e+8 | 8.13e+8 | 8.08e+8 | 8.22e+8 | 8.20e+8 | **8.23e+8** | 0.000 |
| MountainCar | Continuous | 4.53e+6 | 4.61e+6 | 4.50e+6 | 4.80e+6 | 4.64e+6 | 4.97e+6 | 5.00e+6 | **5.02e+6** | 0.000 |
| Water Reservoir | Continuous | 3.12e+5 | 3.24e+5 | 3.06e+5 | 3.08e+5 | 3.07e+5 | 3.18e+5 | 3.42e+5 | **3.44e+5** | 0.000 |
| HopperEnv | Continuous | 6.82e+4 | 8.36e+4 | 6.77e+4 | 9.12e+4 | 6.76e+4 | **9.87e+4** | 9.84e+4 | 9.84e+4 | 0.000 |
| MOHalfcheetah | Continuous | 6.26e+4 | 6.30e+4 | 6.11e+4 | 6.32e+4 | 6.21e+4 | 6.47e+4 | 6.48e+4 | **6.50e+4** | 0.000 |
| MOAnt | Continuous | 1.21e+7 | 1.29e+7 | 1.02e+7 | 1.28e+7 | 1.15e+7 | 1.29e+7 | 1.29e+7 | **1.31e+7** | 0.000 |
| MOSwimmer | Continuous | 1.24e+4 | 1.26e+4 | 9.98e+3 | 1.31e+4 | 1.02e+4 | 1.48e+4 | 1.50e+4 | **1.53e+4** | 0.000 |
| MOHumanoid | Continuous | 2.00e+5 | 1.92e+5 | 1.65e+5 | 2.02e+5 | 1.74e+5 | **2.26e+5** | 2.24e+5 | 2.21e+5 | 0.000 |
| MOWalker2D | Continuous | 5.44e+4 | 5.42e+4 | 5.01e+4 | 5.51e+4 | 5.05e+4 | 5.48e+4 | 5.55e+4 | **5.67e+4** | 0.001 |

## 5.2 RESULTS AND PERFORMANCES

**Hypervolume performances.** Table 1 shows the Hypervolume (HV) values of LacaDM and baseline models in discrete and continuous MORL environments, respectively. The HV metric measures the accuracy and diversity of the solution set, by calculating as the volume of the hypercube between the Pareto Front (PF) and a reference point. We use the default reference points provided by the MOGymnasium framework. A higher HV value indicates better overall algorithm performance.

In Table 1, LacaDM achieved the highest average HV values in five out of eight scenarios in the discrete environments, and in six out of eight scenarios in the continuous environments. The results are based on the average of 10 independent runs, with statistical tests confirming that LacaDM outperformed the baseline models at a significance level of $p < 0.05$ after 10 runs. In the discrete environments, LacaDM particularly excelled in complex scenarios such as Breakable Bottles and MOLunarLander, where it significantly outperformed the baseline models. These results underscore LacaDM's ability to handle environments with intricate reward structures and sparse objective distributions. Similarly, in the continuous environments, LacaDM achieved optimal performance in high-dimensional tasks like MOHalfcheetah, MOSwimmer and MOWalker2D, showcasing its capability to manage the challenges of continuous action spaces and dynamics. The superior HV values achieved by LacaDM across various environments validate its potential for solving complex MORLs. Notably, when compared to MTDiff and DMBP, two reinforcement learning diffusion-based models, LacaDM exhibited clear advantages in both continuous and discrete MORL tasks. This further underscores the effectiveness of our causal representation learning mechanism in enhancing the diffusion process and improving optimization performance across diverse MORL scenarios.

**Sparsity performances.** To further evaluate the performance of our LacaDM, we assess sparsity, which measures how evenly the solutions are distributed across the objective space. Lower sparsity values indicate better performance. The p-values in Table 2 further confirm the statistical significance of these results. As shown in Table 2, LacaDM achieved the lowest average sparsity values in five out of eight discrete environments and six out of eight continuous environments, based on the average of 10 independent runs. These results highlight the effectiveness of LacaDM's diffusion process in generating evenly distributed solutions. Notably, LacaDM performed exceptionally well on high-dimensional continuous tasks, such as MO-Halfcheetah and MO-Walker2D, and complex discrete tasks like MO-Lunar-Lander, where capturing causal dynamics is critical for balancing solution distribution. Overall, LacaDM consistently maintains low sparsity values, demonstrating that causal representation learning, combined with the diffusion process, plays a key role in producing diverse and evenly distributed solution sets, a critical requirement for effective multi-objective optimization.

Table 2: Comparison of average Sparsity across discrete and continuous MORL environments.

| MORL environments | Env. Type | Deep Qlearning | PCN | ANSGAII | NSGAIIIEHVI | EmoDM | MTDiff | DMBP | LacaDM (Ours) | p-value |
|---|---|---|---|---|---|---|---|---|---|---|
| Deep Sea Treasure | Discrete | 1.53e+1 | 1.45e+1 | 2.01e+1 | 1.35e+1 | 1.97e+1 | 1.34e+1 | 1.28e+1 | **1.24e+1** | 0.000 |
| HighwayEnv | Discrete | 1.83e+1 | 1.67e+1 | 2.15e+1 | 1.94e+1 | 2.20e+1 | **1.41e+1** | 1.68e+1 | 1.65e+1 | 0.000 |
| ResourceGathering | Discrete | 8.09e+2 | 6.42e+1 | 1.25e+1 | 8.02e+2 | 1.03e+1 | 6.42e+2 | 6.35e+2 | **6.33e+2** | 0.000 |
| FourRoom | Discrete | 3.37e+2 | 3.42e+2 | 5.21e+2 | 3.30e+2 | 5.17e+2 | 3.18e+2 | **3.11e+2** | 3.15e+2 | 0.000 |
| FruitTree | Discrete | 2.40e+1 | 2.41e+1 | 3.17e+2 | 2.35e+2 | 3.26e+2 | 2.08e+2 | **1.98e+2** | 2.02e+2 | 0.000 |
| BreakableBottles | Discrete | 3.54e+1 | 3.65e+1 | 4.02e+1 | 3.55e+1 | 4.00e+1 | 3.29e+1 | 3.27e+1 | **3.26e+1** | 0.000 |
| Fishwood | Discrete | 1.58e+0 | 1.66e+0 | 2.21e+0 | 1.54e+0 | 2.18e+0 | 1.48e+0 | 1.52e+0 | **1.43e+0** | 0.000 |
| MOLunarLander | Discrete | 1.25e+1 | 1.26e+1 | 1.45e+1 | 1.22e+1 | 1.48e+1 | 1.19e+1 | 1.18e+1 | **1.18e+1** | 0.001 |
| MountainCar | Continuous | 7.57e+1 | 6.12e+1 | 9.04e+1 | 7.51e+1 | 8.97e+1 | 6.21e+1 | 6.18e+1 | **6.02e+1** | 0.000 |
| Water Reservoir | Continuous | 2.14e+0 | 1.95e+0 | 3.45e+0 | 2.01e+0 | 3.33e+0 | **1.88e+0** | 1.94e+0 | 1.92e+0 | 0.000 |
| HopperEnv | Continuous | 1.46e+0 | 1.24e+0 | 3.14e+0 | 1.20e+0 | 3.24e+0 | 1.11e+0 | 1.08e+0 | **1.03e+0** | 0.000 |
| MOHalfcheetah | Continuous | 9.36e+0 | 9.42e+0 | 1.26e+1 | 9.23e+0 | 1.22e+1 | 9.21e+0 | 9.18e+0 | **9.15e+0** | 0.000 |
| MOAnt | Continuous | 2.66e+2 | 2.48e+2 | 3.54e+2 | 2.40e+2 | 3.28e+2 | 2.17e+2 | 2.20e+2 | **2.15e+2** | 0.000 |
| MOSwimmer | Continuous | 9.16e+0 | 9.23e+0 | 1.54e+1 | 9.02e+0 | 1.02e+1 | **8.45e+0** | 8.57e+0 | 8.77e+0 | 0.000 |
| MOHumanoid | Continuous | 5.72e+1 | 5.87e+1 | 6.25e+1 | 5.69e+1 | 6.11e+1 | 5.67e+1 | 5.87e+1 | **5.55e+1** | 0.002 |
| MOWalker2D | Continuous | 2.31e+2 | 2.28e+2 | 2.94e+2 | 2.22e+2 | 2.88e+2 | 2.04e+2 | 2.11e+2 | **2.01e+2** | 0.000 |

**Expected utility maximization performances.** We also use Expected Utility Maximization (EUM) as an evaluation metric. EUM reflects the decision-maker's preferences by calculating a weighted average of utility values across all possible outcomes, where larger values indicate better performance. For each environment, we use the default implementation from the MO-Gymnasium framework to compute EUM values. Figure 2 illustrates the trends of expected utility over training steps for HighwayEnv and Deep Sea Treasure (discrete environments) and MO-Ant and MO-Walker2D (continuous environments). The results consistently show that LacaDM outperforms baseline models across all tasks. In discrete environments, LacaDM demonstrates rapid convergence, achieving higher EUM values than baselines within the first 1000 steps and maintaining its lead throughout training. In continuous environments, LacaDM achieves superior performance by consistently reaching higher EUM values earlier in training. Its performance stabilizes near the optimal solution after 1500 steps, highlighting its ability to efficiently explore and converge in challenging continuous control tasks.

These results reflect LacaDM's advantages in both discrete and continuous settings, where faster convergence and better stability stem from its key mechanisms. Specifically, causal representation learning captures causal relationships between objectives, guiding the reverse diffusion process for more efficient exploration and convergence. Meanwhile, the diffusion process achieves a balance between solution diversity and optimization through the stepwise introduction and removal of noise, enabling LacaDM to generate high-quality strategies more effectively than other methods.

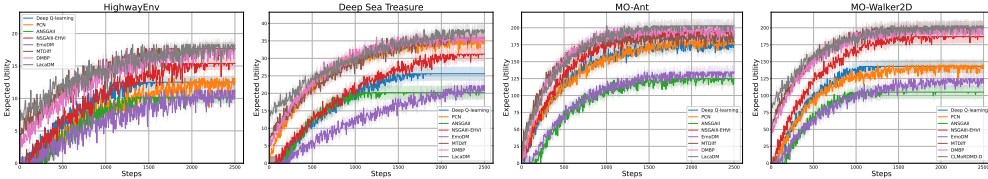

Figure 2: Expected utility of baseline models and LacaDM across four MORL problems as the number of solving steps increases.

## 5.3 Effect of Latent Casual Learning

In this section, we conduct experiments on a variant of LacaDM, LacaDM-CRL (which removes the CRL component), to analyze and validate the effectiveness of causal representation Learning (CRL). To evaluate its performance, we selected four environments: Fishwood and HighwayEnv from the discrete set, and HopperEnv and MountainCar from the continuous set. All other experimental settings remain consistent with the original LacaDM configuration. To isolate the impact of the CRL component in LacaDM, we compared the results of LacaDM with those of LacaDM-CRL. Table 3 presents the comparative results in these four environments, demonstrating the critical role of CRL in the policy generation process. Without CRL, LacaDM loses the ability to capture the potential relationships between the new environment and the predicted noise during strategy generation. Consequently, the reverse diffusion process fails to converge effectively, preventing the model from reaching the optimal strategy.

In addition, to investigate whether CRL is the key factor contributing to this phenomenon, we generated cosine similarity heatmaps of noise inference and training at $\frac{T}{2}$, with and without the CRL component. In our experiment, $T = 1500$, the training environment for the sampled data was MO-SuperMario, and the reasoning environment was MO-Walker2D. The resulting heatmaps are shown in Figure 3. As illustrated in Figure 3, the overall color of the heatmap is noticeably darker when the CRL component is included compared to when it is not. This indicates that our proposed model with the CRL component is better able to generalize from the training environment to an unseen reasoning environment. These results highlight the critical role of CRL in improving the model's ability to adapt and transfer knowledge across different environments.

Table 3: Hypervolume (HV) results of LacaDM with and without the CRL component across four MORL environments.

| Environment | LacaDM-CRL | LacaDM |
|---|---|---|
| Fishwood | 2.42e+3 | **3.15e+3** |
| HighwayEnv | 9.87e+3 | **2.51e+4** |
| MountainCar | 4.24e+6 | **5.02e+6** |
| HopperEnv | 6.38e+4 | **9.84e+4** |

## 6 Conclusion

In this paper, we proposed LacaDM, a novel diffusion model enhanced with CRL to tackle complex MORL problems. LacaDM combines the strengths of diffusion models for effective exploration and convergence with the power of CRL to capture dynamic relationships between objectives and environments. This integration enables LacaDM to produce diverse and high-quality solutions across discrete and continuous MORL environments. Through extensive experiments on MO-Gymnasium environments, we demonstrated that LacaDM consistently outperforms baseline methods , including reinforcement learning algorithms, evolutionary algorithms, and existing diffusion-based approaches, in both discrete and continuous tasks.

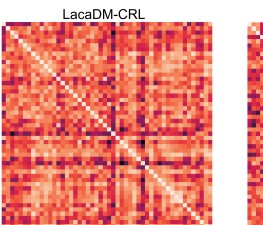

Figure 3: Cosine similarity heatmaps comparing noise inference and training between LacaDM-CRL and LacaDM at the midpoint of the diffusion process.

These results highlight the effectiveness of CRL in improving the diffusion model's ability to generalize across environments and optimize strategies efficiently. In the future, we plan to explore more advanced causal inference techniques for the diffusion model to further enhance the scalability and generalization capabilities of LacaDM. Additionally, we also aim to explore the integration of other advanced techniques such as multi-agent collaboration and transfer learning to further push the boundaries of LacaDM in solving even more complex MORL tasks.

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

# A APPENDIX

## A.1 EXPERIMENT DETAILS

We provide key implementation settings used for training LacaDM across all experiments. These configurations ensure stability, efficiency, and reproducibility of results across different MORL environments. The training was conducted using PyTorch on a single NVIDIA RTX 4090 GPU. All experiments use a fixed random seed of 42 unless otherwise specified. Table 4 summarizes the core training and model parameters.

Table 4: Summary of core implementation configurations.

| Component | Setting | Description |
|---|---|---|
| Optimizer | Adam | Learning rate $1 \times 10^{-4}$ |
| Batch size | 256 | Number of samples per update |
| Epochs | 500 | Full training passes |
| Diffusion steps ($T$) | 100 | Reverse denoising steps |
| Network layers | 3 | Fully connected layers in denoiser |
| Hidden units | 512 | Per layer, with ReLU activations |
| Latent dim ($z$) | 64 | VAE latent space dimension |
| Regularization | $\lambda = 0.01$ | $L_1$ regularization on policies |

## A.2 ADDITIONAL EXPERIMENT RESULTS

### A.2.1 DIFFUSION TIME STEPS ANALYSIS

For LacaDM, the setting of the time step of diffusion is crucial because it is related to the speed of reasoning and the consumption of computing resources. For this reason, we selected three environments, Breakable Bottles, FruitTree and MOHalfcheetah respectively, and drew the curve of HV finger change with the increase of the number of denoising steps in Figure 4.

As observed in Figure 4, the HV stabilizes when the number of steps reaches 1500, indicating that setting the time step to 1500 yields a more optimal tradeoff between performance and computational cost.

### A.2.2 EXPECTED UTILITY MAXIMIZATION PERFORMANCES

Figure 5 shows the change trend of expected utility with the increase of steps when LacaDM and baseline model solve Fishwood, ResourceGathering in discrete environment and MOHalfcheetah, MOHumanoid in continuous environment respectively.

It can be seen from the Figure 5 that although the Expected Utility obtained from LacaDM solution increases slowly in the initial stage, it presents a stable and continuous rising trend with the progress of steps. This shows that the model has effective learning ability and can be continuously optimized and improved in the process. When the step approaches 1500, the LacaDM model becomes stable, indicating that the performance of LacaDM model is gradually stable in the later stage. In addition, the EUM value obtained after the curve converges is better than that of other baseline models.

### A.2.3 EFFECTIVENESS OF THE DIFFUSION PROCESS IN MULTIOBJECTIVE LEARNING

Before evaluating LaCaDM's generalization ability across environments, we first assess whether the diffusion process alone can effectively model multiobjective policies in a single-task setting. To isolate the effect of the diffusion mechanism, we construct a simplified version of LaCaDM by removing the causal module and training and testing within the same environment.

As shown in Table 5, even under this minimal setup, the diffusion-based model consistently outperforms a standard Pareto-conditioned network (PCN) baseline across multiple environments, as measured by hypervolume (HV) and expected utility maximization (EUM). These results demonstrate that the diffusion process alone is sufficient to capture meaningful multiobjective trade-offs, forming a solid foundation for the full model's causal and generalization capabilities.

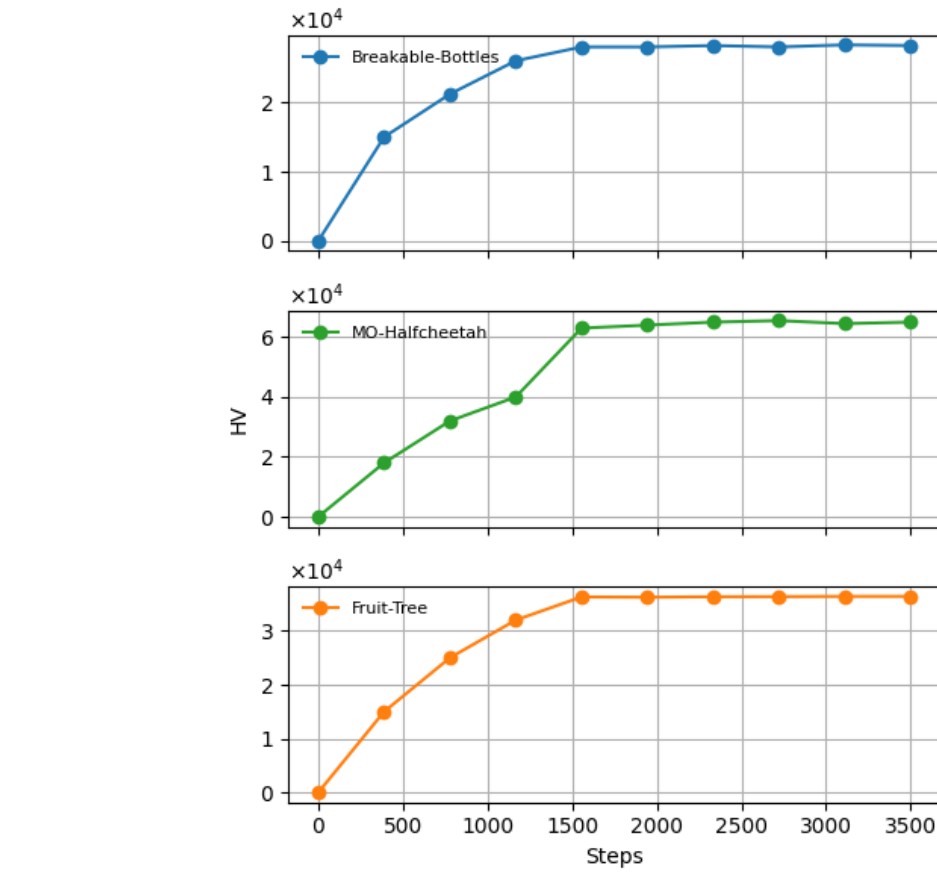

Figure 4: HV indicator versus the number of denoising steps for LacaDM across three distinct environments: Breakable Bottles, MOHalfcheetah and FruitTree.

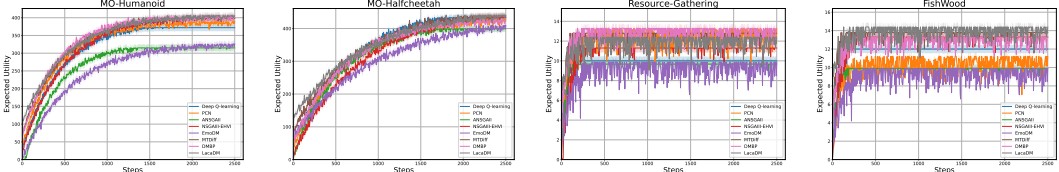

Figure 5: Expected utility vs Steps

## A.3 RATIONALE FOR CHOOSING PCN

We select PCN Reymond et al. (2022) as the reinforcement learning algorithm for generating training data due to its strong alignment with the goals of multiobjective policy modeling. PCN explicitly models Pareto frontiers and adapts to varying preference vectors via conditional networks, enabling the generation of diverse and high-quality policy trajectories. Its conditional mechanism is also well aligned with the stepwise structure of diffusion models, making it suitable for capturing policy evolution over time. Empirically, PCN provides stable and informative trajectories that facilitate effective learning in our diffusion-based framework.

Compared to standard RL methods such as PPO Yu et al. (2022) or SAC Chavali et al. (2022), PCN offers several advantages in multiobjective settings. Traditional RL methods typically optimize scalarized objectives and require retraining when preferences change. In contrast, PCN conditions on preference vectors directly, allowing flexible generation of policies in a single training run Del-

Table 5: Performance of LaCaDM trained and evaluated in the same environment, compared to a Pareto-conditioned network (PCN). Results are averaged over three runs. Higher is better.

| Environment | Baseline (PCN) | | LaCaDM | |
|---|---|---|---|---|
| | HV ↑ | EUM ↑ | HV ↑ | EUM ↑ |
| MOHalfCheetah | 6.30e+4 | 5.10e+3 | **6.42e+4** | **5.65e+3** |
| Fishwood | 3.03e+3 | 6.95e+2 | **3.11e+3** | **7.70e+2** |
| FruitTree | 3.31e+4 | 9.84e+3 | **3.60e+4** | **1.08e+4** |

grange et al. (2023). This improves data efficiency and better supports the conditional generation paradigm required by our diffusion model.

### A.4 RELATED WORK

#### A.4.1 MULTIOBJECTIVE REINFORCEMENT LEARNING AND DIFFUSION MODELS

MORL aims to optimize policies under conflicting objectives, where trade-offs must be learned rather than predefined. Classical approaches, such as scalarization techniques Roijers et al. (2015); Yang et al. (2019), and Pareto-based solutions Reymond et al. (2024); Roijers et al. (2013); Van Moffaert & Nowé (2014), typically require to retrain when objectives or environments change. These methods often assume a static task distribution, limiting their applicability in real-world scenarios with nonstationary dynamics. Recent toolkits such as MOGymnasium Felten et al. (2024a) highlight the need for MORL algorithms capable of adapting to dynamic and uncertain environments. At the same time, diffusion models have gained traction in reinforcement learning due to their ability to generate diverse and coherent action sequences by reversing a noise process Janner et al. (2022); Liu et al. (2024a). While effective in single-objective offline RL and planning, these models typically lack mechanisms for modeling evolving task structures or latent environmental factors Zhang et al. (2024c). Our work addresses this gap by integrating diffusion-based policy generation with latent causal modeling for dynamic MORL.

#### A.4.2 CAUSAL REPRESENTATION LEARNING

Causal representation learning aims to uncover latent factors that govern observable dynamics, offering a principled framework for improving generalization and robustness Liu et al. (2024b); Yao et al. (2021); Tian et al. (2024). Such representations can disentangle exogenous environment shifts from endogenous agent behavior, which is particularly important in nonstationary settings with delayed or indirect effects. In reinforcement learning, causal reasoning supports interventions, counterfactual predictions, and long-term credit assignment Wang et al. (2024).

Although causal modeling has been explored in domains such as domain generalization and imitation learning, its integration into multiobjective RL remains limited Zeng et al. (2024). Most existing works Joshi et al. (2024); Schulte & Poupart (2024) treat causal inference and policy learning as independent components, without leveraging the potential synergy between structured causal modeling and adaptive decision-making. In contrast, our approach embeds a latent temporal causal model directly into the diffusion process, enabling the policy to proactively adapt to shifts in objectives or environment dynamics via causal interventions.

### A.5 DERIVATION OF THE CRL FRAMEWORK

#### A.5.1 PROBLEM FORMULATION

Let $\mathbf{x}_t \in \mathbb{R}^d$ represent the observed temporal data at time $t$. This data is generated by latent causal processes $\mathbf{z}_t \in \mathbb{R}^n$ ($n \leq d$) via an injective mixing function $g : \mathbb{R}^n \to \mathbb{R}^d$:

$$\mathbf{x}_t = g(\mathbf{z}_t) \tag{21}$$

The latent processes $\mathbf{z}_t$ evolve according to causal relationships, where the value of each latent variable $z_t$ at time $t$ depends on the values of its parent variables $\mathbf{Pa}(z_t)$ at the previous time steps,

and a noise term $\epsilon_t$ that introduces uncertainty:

$$z_t = f\left(\mathbf{Pa}(z_t), \epsilon_t\right), \quad \epsilon_t \sim p_{\epsilon_t|\mathbf{u}} \tag{22}$$

where $\mathbf{Pa}(z_t)$ represents the parent variables of $z_t$, and $f$ is a nonparametric function that captures the causal influence of the parents on the current latent variable. The noise $\epsilon_t$ is assumed to follow a distribution modulated by the regime variable $\mathbf{u}$, which may depend on the environment or system dynamics.

### A.5.2 FORWARD DIFFUSION PROCESS

The forward process introduces Gaussian noise over $T$ steps to corrupt the latent variables $\mathbf{z}_t$:

$$q(\mathbf{z}_t|\mathbf{z}_{t-1}) = \mathcal{N}\left(\mathbf{z}_t; \sqrt{1-\beta_t}\mathbf{z}_{t-1}, \beta_t\mathbf{I}\right) \tag{23}$$

The marginal distribution of $\mathbf{z}_t$ after $t$ steps is:

$$q(\mathbf{z}_t|\mathbf{z}_0) = \mathcal{N}\left(\mathbf{z}_t; \sqrt{\bar{\alpha}_t}\mathbf{z}_0, (1-\bar{\alpha}_t)\mathbf{I}\right), \quad \bar{\alpha}_t = \prod_{s=1}^{t}(1-\beta_s) \tag{24}$$

### A.5.3 REVERSE DIFFUSION WITH CAUSAL CONSTRAINTS

The reverse process learns to denoise the latent variables while preserving causal structure:

$$p_\theta(\mathbf{z}_{t-1}|\mathbf{z}_t) = \mathcal{N}\left(\mathbf{z}_{t-1}; \mu_\theta(\mathbf{z}_t, t), \Sigma_\theta(\mathbf{z}_t, t)\right) \tag{25}$$

The latent temporally causal processes Yao et al. (2021) enforces three key constraints:

1. **Independent Noise (IN)**: $\epsilon_{it} \perp\!\!\!\perp \mathbf{Pa}(z_{it})$
2. **Nonstationary Noise**: $p_{\epsilon_{it}|\mathbf{u}}$ varies across regimes $\mathbf{u}$
3. **Sufficient Variability**: For any $\mathbf{z}_t$, $\exists 2n+1$ regimes $\mathbf{u}_j$ such that:

$$\text{rank}\left(\left[\mathbf{w}(\mathbf{z}_t, \mathbf{u}_{j+1})\mathbf{w}(\mathbf{z}_t, \mathbf{u}_j)\right]_{j=0}^{2n}\right) = 2n \tag{26}$$

### A.5.4 OPTIMIZATION FRAMEWORK

The complete objective combines:

$$\mathcal{L}_{\text{ELBO}} = \mathbb{E}_{q_\phi}[\log p_\theta(\mathbf{x}|\mathbf{z})] - D_{\text{KL}}(q_\phi \| p_\psi) \tag{27}$$

$$\mathcal{L}_{\text{TC}} = \mathbb{E}_{\hat{\epsilon}} \log \frac{\mathcal{D}(\hat{\epsilon})}{1 - \mathcal{D}(\hat{\epsilon}^{\text{perm}})} \tag{28}$$

$$\mathcal{L}_{\text{Total}} = \mathcal{L}_{\text{ELBO}} + \lambda\mathcal{L}_{\text{Mask}} + \sigma\mathcal{L}_{\text{TC}} \tag{29}$$

### A.5.5 IDENTIFIABILITY PROOF SKETCH

The latent mapping $h = g^{-1} \circ \hat{g}$ reduces to permutation and componentwise invertible transforms. Key steps:

1. **Observational Equivalence**:

$$p(\mathbf{z}_t|\{\mathbf{z}_{t-\tau}\}, \mathbf{u}) = p(h^{-1}(\mathbf{z}_t)|\{h^{-1}(\mathbf{z}_{t-\tau})\}, \mathbf{u}) \prod_i \left|\frac{\partial h_i^{-1}}{\partial z_{it}}\right| \tag{30}$$

2. **Jacobian Constraints**:

$$\sum_i \mathbf{a}_i q_i^{11} + \mathbf{b}_i q_i^1 = \mathbf{c} \tag{31}$$

3. **Identifiability Result**:

$$h(\mathbf{z}) = \pi \circ T(\mathbf{z}), \quad T(z_i) \text{ is invertible}, \quad \pi \text{ is a permutation} \tag{32}$$

