# OpenReview forum: "A Latent Causal Diffusion Model for Multiobjective Reinforcement Learning"
_ICLR.cc/2026/Conference — ICLR 2026 Conference Withdrawn Submission_

### Official Review · Reviewer_BJqa · 2025-10-29

**Soundness:** 1
**Presentation:** 2
**Contribution:** 2
**Rating:** 2
**Confidence:** 4

**Summary:**

This paper proposes a diffusion model that utilizes causal representation learning to enhance performance and generalization in diverse multi-objective reinforcement learning tasks. Empirical evaluations are presented to demonstrate the performance of the proposed model.

**Strengths:**

1. This work investigates the potential of combining causal representation learning to enhance the performance and generalization of diffusion models for MORL.
2. Both continuous and discrete action spaces are taken into consideration, which is comprehensive.

**Weaknesses:**

1. All numerical results are presented without standard errors or confidence intervals, making it hard to interpret the results.
2. The curves in Figure 2 and Figure 5 seem suspicious, as many of them converge and become fixed in both average and standard error. Explanations are needed.
3. This work claims that LacaDM can “adapt” to dynamic environments and “continuously learn.” However, no experimental results support these claims.
4. The results in Figure 3 are poorly explained, and it is hard to understand how CRL and full LacaDM promote generalization.
5. It would be better to list the definitions of the three metrics (HV, SP, EUM) to help readers understand the results.
6. In Section 2.2, MORL environments are typically modeled as multi-objective MDPs (MOMDPs) instead of MDPs.
7. As the proposed model focuses on learning from pre-collected trajectory datasets, there should be experiments investigating the influence of dataset quality.

**Questions:**

1. How does causal representation enhance generalization? An example would be appreciated.
2. How is the problem of distribution shift solved, as the model is learning from offline datasets?
3. Can the diffusion steps be improved?
4. In the results of Figure 4, it requires nearly 1500 diffusion steps to maximize HV. Why are the diffusion steps set to 100 as listed in Table 4? Would that be enough?
5. How does the derivation of the CRL framework proceed in Appendix A.5? I cannot follow the arguments, as they only list many concepts and definitions.

---

### Official Review · Reviewer_5y78 · 2025-10-29

**Soundness:** 3
**Presentation:** 2
**Contribution:** 2
**Rating:** 2
**Confidence:** 4

**Summary:**

### Summary

This paper proposes a novel method, LacaDM, to address challenges in multi objective reinforcement learning, particularly objective conflicts and poor adaptability in dynamic environments. LacaDM integrates CRL with diffusion models, aiming to learn the latent temporal causal relationships between environmental states and policies.

**Strengths:**

### Strengths

1.The experimental design is comprehensive, covering a diverse set of 8 discrete and 8 continuous environments.

2.The empirical results demonstrate that LacaDM significantly outperforms the baselines on most tasks, showing clear advantages in both Hypervolume and Sparsity metrics.

3.The integration of concepts from causal reinforcement learning is a novel contribution.

**Weaknesses:**

### Weaknesses

1.The manuscript lacks clarity, leaving several key settings ambiguous. Most importantly, it is unclear whether the paper addresses an offline or online multi objective reinforcement learning problem. Based on the experimental setup, which involves pre collecting data for training, the method appears to be designed for the offline setting.

2.If the paper targets the offline setting, several core motivations seem contradictory. For instance, the paper states in lines 65-68 that existing models "fail to account for the temporal and latent dependencies that arise from the interaction between the agent’s actions and the evolving environment". In an offline setting, the environment is static by definition. Why then is the challenge of an "evolving environment" emphasized as a motivation?

3.Offline multi objective reinforcement learning is an established research area featuring well known benchmarks and baselines, such as Pareto Efficient Decision Agents PEDA[1]. Furthermore, several existing works combine diffusion models with multi objective reinforcement learning [2, 3], yet these are not discussed or compared. The baseline MTDiff, which is a multi task diffusion model, does not seem to be a closely matched comparison for this specific problem.

4.The calculation methods for Hypervolume and Sparsity are not detailed. It is also unclear what "10 independent runs" signifies. Does this imply sampling only 10 preference vectors? This seems low compared to benchmarks like PEDA, which typically sample 255 preferences for comprehensive coverage.

5.Several definitions are ambiguous. It is unclear whether pi represents a parameterized policy or a specific action. Does the diffusion model generate sequences of actions or policy network parameters? For example, pi is defined as a policy, but in Equation 15, it appears to represent an action sequence. Furthermore, the paper fails to distinguish between the diffusion time steps and the environment's decision time steps, which is a critical distinction in diffusion based decision making.

6.There is a minor typo: the term should be "Multi Objective".

[1] Scaling Pareto-Efficient Decision Making Via Offline Multi-Objective RL.  ICLR2024
[2] MODULI: Unlocking Preference Generalization via Diffusion Models for Offline Multi-Objective Reinforcement Learning. ICML2025
[3] Generalizable Offline Multiobjective Reinforcement Learning via Preference-Conditioned Diffuser. TNNLS

**Questions:**

see weakness.

---

### Official Review · Reviewer_WU4c · 2025-10-30

**Soundness:** 2
**Presentation:** 2
**Contribution:** 2
**Rating:** 4
**Confidence:** 2

**Summary:**

This paper proposes LacaDM, which utilizes Reinforcement Learning (RL) context embeddings to guide a diffusion process for generating an optimal policy to solve Multi-Objective Reinforcement Learning (MORL) problems. The RL context embeddings are derived from PCN-generated trajectories, and the latent variables $z$ are learned via CRL.

**Strengths:**

The proposed method is intriguing and demonstrates strong performance when compared to both continuous and discrete MORL algorithms.

**Weaknesses:**

- The paper is extremely difficult to follow. The notation is confusing and inconsistent, making the methodology hard to grasp starting from Section 4 (Methodology). (Detailed issues are listed in the Questions section below).

- The algorithm's name is inconsistent, appearing as "LaCaDM" in some parts of the paper and "LacaDM" in others. This should be standardized.

**Questions:**

There are several critical ambiguities in the description of the diffusion process and the meaning of the policy $\pi$: Line 247, Page 5: The text states, "We begin by solving $N$ distinct MORL problems to convergence and recording the resulting policy $\pi_T$ and cumulative reward sequences $R_T$." $T$ is defined as the total number of time steps in the diffusion process. Why does solving $N$ distinct MORL problems yield $T$ policies ($\pi_T$)? How are the noisy policies $\pi_t$ for $t < T$ obtained from these $N$ converged solutions? In Section 2.2, $\pi$ is defined as a mapping from state to action. In the context of RL practice, this mapping is typically parameterized by a neural network. What exactly does the "policy generated by the diffusion process" refer to? Does it represent the entire set of neural network parameters? What is the neural network architecture of the diffusion model? How is it trained to output the complete set of neural network parameters that define the policy mapping $\pi(s)$? The definition of $R_t$ is unclear. The preceding definitions only introduce the vector reward $r$ and the vector cumulative return $G$. What is the scalar quantity $R_t$ defined here? Furthermore, how are $T$ different cumulative rewards generated from solving $N$ problems?

---

### Official Review · Reviewer_PJ2j · 2025-10-30

**Soundness:** 2
**Presentation:** 2
**Contribution:** 3
**Rating:** 2
**Confidence:** 4

**Summary:**

This paper proposes LacaDM, a latent causal diffusion model for multi-objective reinforcement learning (MORL). After generating trajectories using a trained MORL algorithm (e.g., PCN in this case), the diffusion model is trained to reproduce the policy information embedded in those trajectories. In addition, a latent representation is learned through an encoder-decoder structure that predicts denoised samples, ensuring causal consistency between latent variables and policy evolution. During deployment, the causality-aware generative model is used for policy generation. Experimental results show that LacaDM outperforms baseline methods across standard MORL metrics.

**Strengths:**

- Provides a clear and detailed explanation of the background.

- Tackling representation learning to improve generalization and model transferability in MORL is a promising research direction.

**Weaknesses:**

1. Flow of the proposed algorithm is hard to understand. Several key components should be clarified.
- Please provide a pseudocode for better readability.
- Section 4.3 – the core part of this paper – should be clarified.
   - It seems that \hat{pi}_t and \hat{pi}_{t-1} should be interchanged in Eqs. (17), (18), and the second term in Eq. (19) because the denoising process is considered. Plus, Section 4.3 is inconsistent with the notation in Fig. 1: pi_0 is random policy in Fig. 1 while it is the optimal one in Section 4.3 (Line 272).
   - What is \hat{pi}_0? Is it the output of the policy, the single feature vector of the trained PCN network, or the PCN network itself (i.e., multiple layers)? If the latter is the case here, the proposed method may have low flexibility because the model configuration is fixed in advance. (I guess the first one may be used here because then we can apply discrete diffusion in the discrete action envs.)
   - Which of the following is used for generating the denoised \hat{pi}_{t-1}? (a) Generate it using Eq. (17), and then z_t is separately trained following (18). (b) Generate \hat{pi}_{t-1} using Eq. (17) conditioned on z_t. I guess the first one is used here because it seems that z_t is separately trained in the second term of Eq. (19).
   - What would be the advantage of the proposed method using diffusion model for “policy” generation compared to using diffusion model for action (sequence) generation?
- How does the inference/deployment phase look like? For example, after training the model in Figure 1, do we generate some policies \hat{pi}_{0} (either feature or PCN model itself) and the corresponding latent vectors \{ z_{t-\tau} \} and use them as in Eq.(13)? (This will be explained once a pseudocode is provided.) Do we further train $\pi_\theta$ in Eq. (13)?

&nbsp;

2. More ablation studies should be conducted.
-       Ablation on the effect of beta and lambda in Eq. (19) is required.
-	How LacaDM-CRL is implemented? Just setting $\beta=0$ in Eq. (19) or no usage of $\{ z_{t-\tau} \}$ in Eq. (13)? Please clarify this.
-	Why making \hat{pi} sparse is useful in Eq. (19)?
-	Please clarify the color using explicit color bar in Fig. 3.
-	How does the final policy feature (output of the penultimate layer) look like? Please visualize w/ and w/o CRL (e.g., using tSNE).
-	Can we ablate the effect of diffusion? i.e., What if we consider CRL concept + MORL without diffusion?

&nbsp;

3. Comparison with other literature in MORL is required.
-	It seems that standard MORL algorithms are missing in the experiment part – Envelope Q-learning, CAPQL, Q-Pensieve, and C-MORL (Liu et al., 2025b).
-	Literature review should be explicitly included in the main paper.
-	Is this work the first one using diffusion in MORL?

&nbsp;

4. Discussion on the computation of the proposed algorithm is required - algorithmic complexity and/or wall-clock time – because diffusion model may cause huge computation complexity/time.

&nbsp;

5. Please clarify the details regarding trajectory generation using PCN.
-	Any reason for choice of the 4 envs? What if we try less or more env?
-	Can we use other MORL algorithm than PCN for trajectory generation?
-	How many trajectories are generated?
-	How h (desired horizon in PCN) is chosen?

**Questions:**

Please see the Weaknesses above. Here are some minor issues.

-	Please use citep appropriately.
-	Section 2.1: binary case only in eq. (5), (6)?
-	Multiobjective  -> Multi-objective
-	MDP -> MOMDP
-	Please use acronym (RL, MORL) appropriately.
-	Please provide code if available.

---

### Note · Authors · 2025-12-02

I have read and agree with the venue's withdrawal policy on behalf of myself and my co-authors.